# Statistical Floquet prethermalization of the Bose-Hubbard model

Emanuele G. Dalla Torre[1,2] and David Dentelski[1,2]

[1]*Department of Physics, Bar-Ilan University, Ramat Gan 5290002, Israel*
[2]*Center for Quantum Entanglement Science and Technology,*
*Bar-Ilan University, Ramat Gan 5290002, Israel*

(Dated: May 28, 2021)

The manipulation of many-body systems often involves time-dependent forces that cause unwanted heating. One strategy to suppress heating is to use time-periodic (Floquet) forces at large driving frequencies. For quantum spin systems with bounded spectra, it was shown rigorously that the heating rate is exponentially small in the driving frequency. Recently, the exponential suppression of heating has also been observed in an experiment with ultracold atoms, realizing a periodically driven Bose-Hubbard model. This model has an unbounded spectrum and, hence, is beyond the reach of previous theoretical approaches. Here, we study this model with two semiclassical approaches valid, respectively, at large and weak interaction strengths. In both limits, we compute the heating rates by studying the statistical probability to encounter a many-body resonance, and obtain a quantitative agreement with the exact diagonalization of the quantum model. Our approach demonstrates the relevance of statistical arguments to Floquet perthermalization of interacting many-body quantum systems.

The study of periodically driven systems has a long history, tracing back to the work of Floquet on classical systems governed by linear equations of motion [1]. Floquet showed that these equations can be solved using a time-independent unitary matrix, $U_F$, which captures the evolution over one period of the drive, $\tau$. Remarkably, because the time evolution of quantum systems is determined by a linear equation (namely, the Schrödinger equation), Floquet theory can be used to study any quantum system, even in the presence of interactions. The practical applicability of Floquet theory is hindered by the fact that finding $U_F$, and diagonalizing it, is generically very difficult. This difficulty is especially acute for many-body quantum systems, where the size of $U_F$ grows exponentially with the number of degrees of freedom. Nevertheless, at large driving frequencies, $U_F$ can be derived using a controlled analytical approximation, the Magnus expansion [2]. The first term of this expansion is $U_F \approx e^{-iH_{av}\tau}$, where $H_{av} = \tau^{-1}\int_0^\tau H(t)dt$ is the time-averaged Hamiltonian. The other terms are integrals of commutation relations of the Hamiltonian at different times [3].

Using the Magnus expansion, Refs. [4–8] were able to obtain *rigorous* constraints on the time evolution of periodically driven quantum many-body systems. These rigorous theorems apply to quantum spin systems that satisfy a local norm bound: their Hamiltonians consist of sums of local operators whose matrix elements are smaller than a given energy scale $J$. For these systems, the heating rate $\Phi$ was shown to be exponential suppressed at large driving frequencies $\Omega = 2\pi/\tau$, according to

$$\Phi(\Omega) < \frac{AJ}{\hbar}\exp\left(-\frac{\hbar\Omega}{BJ}\right), \qquad (1)$$

where $\hbar$ is the Plank's constant, $A$ and $B$ are unitless constant. This exponential suppression was observed in

several numerical studies [9–12] and in an experiment with dipolar spin chains [13].

The rigorous bound of Eq. (1) can be understood using a perturbative argument [4]: Due to the local norm bound, a single application of the driving field can change the energy of the system by $J$, at most. On the other hand, the absorption of a quantum of energy from the pump injects energy $\hbar\Omega$. Hence, the absorption of energy from the pump requires the product of $n = \hbar\Omega/J$ operators and is governed by the $n$th order perturbation theory. Refs. [5–8] used the Magnus expansion to extend this argument and demonstrate that Eq. (1) is a rigorous bound, valid to all orders. Interestingly, in the limit of $\hbar \to 0$, this bound applies to classical systems with a bounded spectrum [14, 15].

Many physical systems escape the regime of validity of the aforementioned rigorous bounds. For example, massive particles with momentum $p$ have a kinetic energy $p^2/2m$ that is unbounded from above. Ref. [16] demonstrated that systems of interacting particles can, nevertheless, show an exponential suppression of heating. They considered a canonical model of coupled kicked rotors [17–20] and showed that, for appropriate initial conditions, the system shows an exponentially long-lived prethermal plateau with vanishing energy absorption. This effect was explained in Ref. [21] using the following statistical argument: At large driving frequencies, the heating rate is small and the time-averaged energy of the system is (quasi) conserved. If the system is ergodic, the state of the system can be approximated by the Boltzmann distribution function,

$$P = Z\exp\left(-\frac{H_{av}}{k_BT}\right), \qquad (2)$$

where $Z$ is the partition function, $k_B$ is the Boltzmann constant, and the temperature $T$ is determined by the initial energy of the system, measured with respect to the

time-averaged Hamiltonian $H_{av}$. If other quantities, such as the total momentum or the total number of particles are conserved, the appropriate Lagrange multipliers need to be taken into account. The resulting distribution can then be used to estimate the heating rate by computing the probability to incur into a many-body resonance [22]. Under physical assumptions, this probability is exponentially small, leading to a *statistical* Floquet prethermalization [21].

Having introduced the concepts of rigorous and statistical Floquet prethermalization, we now move to the focus of this article, namely the periodically driven Bose-Hubbard model, described by

$$H(t) = \frac{U}{2} \sum_i n_i^2 - J(t) \sum_{\langle i,j \rangle} \left( b_i^\dagger b_j + H.c \right), \qquad (3)$$

with $J(t) = J_0 + \delta J \cos(\Omega t)$. Here, $b_i$ and $b_i^\dagger$ are canonical bosonic operators, $n_i = b_i^\dagger b_i$ is the number of particles on site $i$ and $\langle i, j \rangle$ are nearest neighbors. The $U$ term describes onsite repulsion and the $J$ term hopping. Importantly, the $U$ term is unbounded from above, making the rigorous bounds of Ref. [4–8] unapplicable. The Hamiltonian of Eq. (3) conserves the total number of particles in the system, $N = \sum_i n_i$ and $\bar{n}$ denotes the average number of particles per site.

Floquet prethermalization in the Bose-Hubbard model was studied theoretically in Ref. [23] using a self-consistent quadratic approximation. This work employed the concept of many-body parametric resonance [24] to predict the existence of a frequency threshold above which the system does not absorb energy. However, in practice, terms that are neglected in the quadratic approximation lead to finite heating rates at all frequencies. Ref. [4] predicted that at large driving frequency, the heating rate should be rigorously bounded by a stretched exponential [25]. In the limit of a large number of particles per site ($\bar{n} \gg 1$), the model can be mapped to a system of classical rotors, where the heating rate is exponential suppressed [21].

Recently, the heating rate of the Bose-Hubbard model with one particle per site ($\bar{n} = 1$) was studied by Ref. [26], using three methods: (i) the numerical calculation of the linear response of the model; (ii) the experimental measurement of single-site excitations (doublons or holes); (iii) the experimental measurement of the system's temperature. The experiments were performed using ultracold atoms in one and two-dimensional optical lattices. The time-periodic drive was obtained by modulating the intensity of the laser fields that generate the lattice [27]. The findings of Ref. [26] demonstrate that the heating rate is exponentially suppressed as a function of $\Omega$ in all dimensions. As explained, this observation cannot be accounted by the available theoretical methods.

In this article, we present two semiclassical approximations that capture the exponential suppression of the heating in two opposite limits. The first limit is strong interactions ($U \gg J$), where we link the heating suppression to the low probability of finding many particles on a single site. The second limit is weak interactions ($U \ll J$), where we can perform a controlled expansion of the heating rate in orders of $U$. For both cases, we use a statistical approach to compute the heating rate to lowest order in the strength of the periodic drive ($\sim \delta J^2$) and compare it with the exact numerical diagonalization of the model.

**Strong interactions ($U \gg J$)** – In the regime of large interactions, $U \gg J$, we can describe the system in terms of semiclassical particles hopping on a lattice. The periodic drive moves one particle from one site to a neighboring one. This process changes the value of the on-site interaction by

$$\Delta E = \frac{U}{2} \left[ (n_i \pm 1)^2 + (n_j \mp 1)^2 \right] - \frac{U}{2} \left[ (n_i)^2 + (n_j)^2 \right]$$
$$= U[\pm(n_i - n_j) + 1], \qquad (4)$$

where the upper (or lower) sign refers to a particle hopping from site $j$ to site $i$ (or *vice versa*). Following Ref. [21], we need to identify the many-body resonances of the model. Here, a resonance occurs when Eq. (4) equals to an integer multiple of the frequency of the drive (in units of Schrödinger's equation constant $\hbar$), or $\Delta E = m\hbar\Omega$, where $m$ is an integer. For high-frequency drives, the heating rate is dominated by the lowest-order available resonance, which corresponds to $m = \pm 1$. Without loss of generality, we assume that $n_i > n_j$, such that when a particles moves from $j$ to $i$ (or *vice versa*) the interaction energy increase (decreases). The resonance condition $\Delta E = \pm\hbar\Omega$ becomes $\pm(n_i - n_j) + 1 = \pm n_\Omega$, or

$$n_j = n_i - n_\Omega \pm 1. \qquad (5)$$

where we defined $n_\Omega = \hbar\Omega/U$. Here, the upper (or lower) sign refers to the absorption (or emission) of energy. Note that this condition can be matched only if $n_\Omega$ is integer. If the maximal occupation of each site is limited to $n_i \leq 2$, such as in the case of spin-1/2 fermions, the resonant condition can be satisfied only for $n_\Omega = 1$ [28]. In contrast, for bosons $n_i$ is unbounded and energy can be resonantly absorbed at arbitrarily high frequencies. Because the probability to find sites with large $n_i$ is exponentially small, so is the probability to satisfy the resonance condition, leading to suppressed heating rates. The goal of this article is to put this intuitive argument on solid mathematical ground.

The probability to satisfy Eq. (5) is determined by $P_{i,j}(n_i, n_j)$, the joint distribution function to find $n_i$ and $n_j$ particles in sites $i$ and $j$, according to

$$P_\pm(\Omega) = \sum_n P_{i,j} \left( n, \ n - n_\Omega \pm 1 \right). \qquad (6)$$

This expression needs to be multiplied by a factor of 2 to take into account the case of $n_i < n_j$. In a $d$ dimensional

square lattice, we need to further multiply the result by the coordination number $d$ [29].

Evaluating the distribution function $P_{i,j}(n_i, n_j)$ in a (pre)thermal state described by Eq. (2) is a formidable task in many-body quantum physics. In what follows, we focus on the regime of large temperatures $T \gg J$, where we can neglect quantum fluctuations and describe the prethermal state by

$$P_{i,j}(n_i, n_j) = P_i(n_i)P_j(n_j), \tag{7}$$

with

$$P_i(n) = P_j(n) = Z_0 \exp\left(-\frac{U}{2k_BT}n^2 - \frac{\mu}{k_BT}n\right). \tag{8}$$

Here, in addition to the quasi-conservation of the energy in the prethermal state, we took into consideration the conservation of the total number of particles, through the chemical potential $\mu$. The values of $Z_0$ and $\mu$ are determined by the constraints $\sum_n Z_i(n) = 1$ and $\sum_n nZ_i(n) = \bar{n}$.

These constraints, along with the numerical solution of Eqs. (6)-(8) enable us to compute the semiclassical heating rate of the Bose-Hubbard model, $\Phi$. The total heating rate is given by the probability to incur into a resonance ($P_+ - P_-$), times the heating rate of an individual resonance. According to the linear response theory, one obtains

$$\hbar\Phi(\Omega) = (\delta J)^2 (P_+ - P_-)\delta(\hbar\Omega - \Delta E), \tag{9}$$

where the delta function $\delta(\hbar\Omega - \Delta E)$ imposes the relevant resonance condition. To regularize this function, one needs to take into account the effects of small, but finite, $J/U$: the hopping term in Eq. (3) transforms the single particle states into "conduction bands" of width $\Lambda = 4dJ$. To model this effect, we substitute the delta function in Eq. (9) by a square function of width $2\Lambda$, namely $\delta(\hbar\omega) = [\Theta(\hbar\omega > -\Lambda) - \Theta(\hbar\omega > \Lambda)]/(2\Lambda)$, where $\Theta$ is the Heaviside function. In Fig. 1, we plot the resulting heating rates in $d = 1$, obtained from the numerical solution of our semiclassical approach, Eq. (6)-(9), for different values of the temperature [30]. We find that the heating rate is exponentially suppressed for all temperatures and, at large temperatures, inversely proportional to the temperature.

To gain physical insight into this result, we now develop an analytical high-temperature expansion. In the limit of $T \to \infty$, the distribution function is solely determined by the conservation laws and

$$P_i(n) = Z_0 \exp\left(-\frac{\mu n}{k_BT}\right) \equiv Z_0 z^n \tag{10}$$

with $Z_0 = 1 - z$ and $z = \bar{n}/(1 + \bar{n})$ [31]. By combining Eqs. (6) and (10), we obtain

$$P_+ = (1 - z)^2 \sum_{n=n_\Omega}^{\infty} z^{2n-n_\Omega+1} = \frac{1-z}{1+z}z^{\hbar\Omega/U+1} \tag{11}$$

$$P_- = (1 - z)^2 \sum_{n=n_\Omega+1}^{\infty} z^{2n-n_\Omega-1} = \frac{1-z}{1+z}z^{\hbar\Omega/U+1}. \tag{12}$$

Note that the two sums have different lower limits because $P_+$ can occur only if $n_j \geq 1$, while $P_-$ requires only $n_j \geq 0$. Because $P_+ = P_-$ the net energy absorption is zero, $\Phi = 0$. This result is not surprising: infinite temperature ensembles do not absorb energy!

We can use this result as the starting point of a perturbative analysis. By approximating Eq. (8) as $P \approx Z_0\left(1 - Un^2/(2k_BT)\right)e^{-\mu n}$ [32] we obtain

$$P_\pm = Z_0^2 \sum_{n_i-n_j=n_\Omega\pm1} \left[1 - \frac{U}{2k_BT}(n_i^2 + n_j^2)\right]z^{n_i}z^{n_j}, \tag{13}$$

leading to (see symbolic script in appendix B)

$$P_+ - P_- = \frac{\hbar\Omega}{k_BT}\frac{1-z}{1+z}z^{\hbar\Omega/U+1}. \tag{14}$$

In particular, at $\bar{n} = 1$ ($z = 1/2$), we obtain

$$\Phi(\Omega) = \frac{(\delta J)^2\Omega}{24Jk_BT}\exp\left(-\log(2)\frac{\hbar\Omega}{U}\right). \tag{15}$$

Eq. (15) shows that, in the regime of $U \gg J$ and at very high temperatures, the heating rate of the Bose-Hubbard model is an exponential function of the ratio between the driving frequency and the onsite interaction. At intermediate temperatures, the heating rate is additionally suppressed by the fact $Un^2/(2k_BT)$ in Eq. (8), leading to a faster-than-exponential decay of $\Phi(\Omega)$, see Fig. (1). Hence, Eq. (15) can be considered as an upper bound of the heating rate at all temperatures.

We now compare the results of our semiclassical approximation with the exact diagonalization of the Bose-Hubbard model. At finite temperatures, linear response gives [26]

$$\hbar\Phi(\Omega) = \frac{\delta J^2}{2L}\sum_{m,n}|\langle\psi_n|V|\psi_m\rangle|^2\,\delta(E_n - E_m - \hbar\Omega)$$
$$\times \frac{1}{Z}\left(e^{-E_m/k_BT} - e^{-E_n/k_BT}\right). \tag{16}$$

Here $|\psi_n\rangle$ and $E_n$ are, respectively, the eigenstates and eigenvalues of the average Hamiltonian $H_{\text{av}}$ at $\bar{n} = 1$ and $V = \sum_{\langle i,j\rangle} b_i^\dagger b_j + \text{H.c.}$ is the time-dependent perturbation. We evaluate this quantity numerically for $N = 9$ particles on a one dimensional lattice with $L = 9$ sites ($\bar{n} = N/L = 1$) and open boundary conditions [33]. To mitigate the effects due to the finite dimension of the lattice, we have regularized the delta function of Eq. (16)

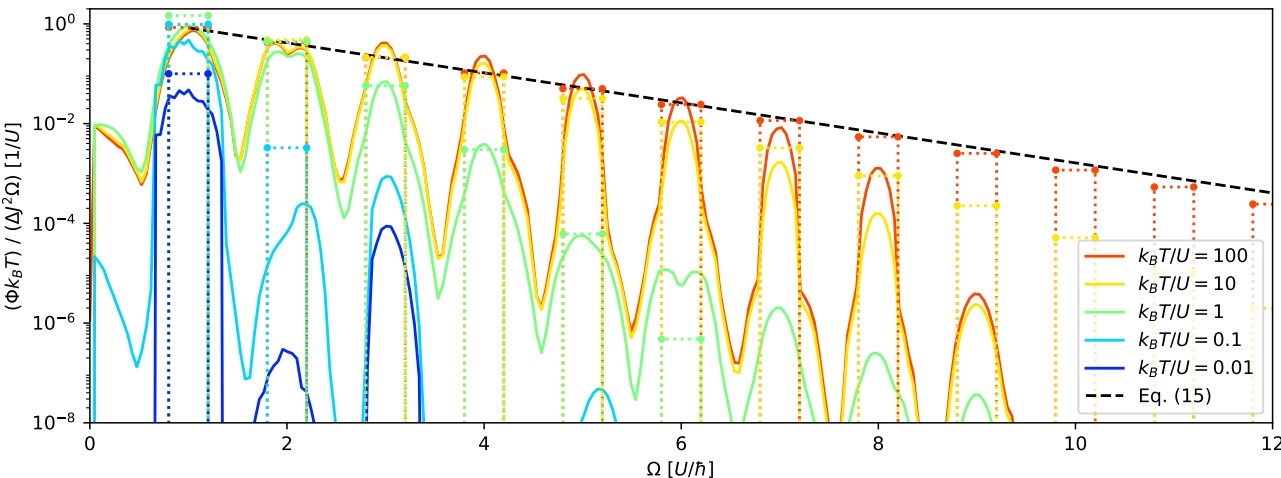

FIG. 1. Heating rate of the Bose-Hubbard model at $\bar{n} = 1$ for $J/U = 0.05$: (i) High-temperature expansion, Eq. (15 (dashed line); (ii) Semiclassical approximation based on Eqs. (6) - (8) (dotted lines); (iii) Exact diagonalization of $N = 9$ particles on $L = 9$ sites (continuous lines).

using the above-mentioned square function with $\Lambda = 2J$. Because the maximal number of particles per site is always smaller or equal to the total number of particles $N$, we need to restrict ourselves to frequencies $\Omega$, such that $n_\Omega < N$, or $\hbar\Omega < NU$ [34]. As shown in the Fig. 1, for all temperatures $T > U$ the results of our numerical calculations are well approximated by the semiclassical description.

**Weak interactions** $(U \ll J)$ – We now turn to the other extreme limit, where the interactions are small in comparison to the kinetic energy and can be treated perturbatively. The periodically driven Bose-Hubbard model of Eq. (3) can be written as the sum of a time-independent part $H = H_0 + H_{\text{int}}$ and a periodic drive, $\delta J \cos(\Omega t) V$, with

$$H_0 = \sum_k (\varepsilon_k - \mu) b_k^\dagger b_k = \sum_k \xi_k b_k^\dagger b_k, \quad (17)$$

$$H_{\text{int}} = \frac{U}{2} \sum_{p',k',p} b_{p'}^\dagger b_{k'} b_p^\dagger b_{p+p'-k'},$$

$V = \sum_k b_k^\dagger b_k$, and (in $d = 1$) $\varepsilon_k = 2J_0 [1 - \cos(k)]$, $b_k = L^{-1/2} \sum_x e^{ikx} b_x$. Using this notation, the heating rate of Eq. (9) takes the form

$$\hbar\Phi(\Omega) = \frac{(\delta J)^2}{2\hbar L} \int_0^\infty d\tau \, e^{-i\Omega\tau} \langle [V(t+\tau), V(t)] \rangle_T, \quad (18)$$

where the square brackets denote a commutator and $\langle ... \rangle_T$ is the expectation value with respect to a thermal state of the time-independent Hamiltonian $H_0 + H_{\text{int}}$ at temperature $T$. This expression can be computed numerically using path integrals techniques, either as the analytic continuation of an imaginary-time correlator, or as a real-time (Keldysh) response function.

In what follows, we present a semiclassical approach, aimed at computing $\Phi$ for $U \ll J$. As we will show below, our approach captures the correct scaling laws of $\Phi$ and highlights its exponential suppression at large frequencies. We treat the eigenstates of $H_0$ as classical particles (quasi-particles), generated by the interaction term $H_{\text{int}}$. The probability to observe a process involving the $n$th order of $H_{\text{int}}$ is given by

$$P(n) = \frac{1}{n!} \left( \frac{U}{J} \right)^n. \quad (19)$$

Here, the factor $n!$ derives from the $n$th order Taylor expansion of the exponent used in the perturbation theory. At zero temperature, this process creates up to $n_{qp} = n/2 + 1$ quasiparticles. This relation is justified by the diagrams shown in Fig. 2, which demonstrate that the leading order contribution to the creation of $n_{qp}$ quasi-particles involves $n = 2n_{qp} - 2$ vertexes. From a semiclassical perspective, this relation indicates that the second order perturbation creates two quasiparticles ($n_{qp} = 2$ for $n = 2$), and that the number of quasiparticles increases by one for every two additional orders of perturbation.

We now use the statistical approach of Eq. (9) to compute the heating rate. A many-body resonance condition is satisfied when the total energy is conserved, namely if $\hbar\Omega = \sum_{j=1}^{n_{qp}} \varepsilon_{k_j} < 4Jn_{qp}$. Hence, the lowest order resonance is obtained for $n_{qp}^* = \lceil \Omega/4J \rceil$, where $\lceil ... \rceil$ is the ceil function. The heating rate is, then, given by $\hbar\Phi_{T=0} = (\delta J)^2 P(n)/J$, or

$$\hbar\Phi_{T=0}(\Omega) = \frac{(\delta J)^2}{J(2n_{qp}^* - 2)!} \left( \frac{U}{J} \right)^{2n_{qp}^* - 2}. \quad (20)$$

In Fig. 3(a) we compute $\Phi_{T=0}(\Omega)$ as a function of $U/J$

$n = 2, \quad n_q = 2$  $n = 4, \quad n_q = 3$

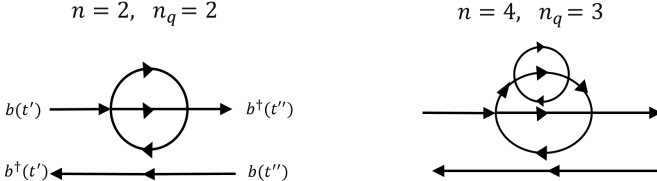

$b(t') \longrightarrow$ $b^\dagger(t'')$

$b^\dagger(t') \longleftarrow$ $b(t'')$

FIG. 2. Representative diagrams for the leading contributions in the second ($n = 2$) and fourth ($n = 4$) orders of the perturbative expansion. The annihilation (creation) operators are denoted by outgoing (incoming) arrows, from left to right. The maximal number of simultaneous quasiparticles is $n_{qp} = 2$ and $n_{qp} = 3$ for the second and fourth orders, respectively. All other diagrams of the same order (i.e. with the same number of vertexes) create less quasiparticles. The generalization to higher order diagrams is straightforward and shows that the $n$th-order perturbation can create up to $n_{qp} = n/2 + 1$ quasiparticles.

using the exact diagonalization of a finite-size system ($L = N = 9$) and show that Eq. (20) captures the correct scaling behavior. As one increases the driving frequency $\Omega$, the heating rate is dominated by higher orders of perturbation theory in $U/J$. Hence, at a fixed $U/J < 1$, the heating rate decreases exponentially with $\Omega$. To see this effect, we consider a smooth version of Eq. (20) by approximating $\lceil x \rceil \approx x + 1/2$ and substituting $n! \to \Gamma(n)$, leading to

$$\hbar\Phi(\Omega) = \frac{(\delta J)^2}{J\Gamma(\hbar\Omega/2J - 1)} \left(\frac{U}{J}\right)^{\frac{\hbar\Omega}{2J} - 1}. \quad (21)$$

This expression is found to be in quantitative agreement with the exact diagonalization calculations, see Fig. 3(b).

We now study the temperature dependence of the heating rate by considering the statistical properties of the aforementioned semiclassical quasiparticles. For simplicity, we approximate the band structure $\varepsilon(k)$ as two plateaus, one at $\varepsilon_{k=0} = -2J$ and one at $\varepsilon_{k=\pi} = 2J$. In this simplified model, the creation of a quasiparticle involves an energy jump of $\Delta\varepsilon = 4J$. This event is possible only if the $k = 0$ state is full and the $k = \pi$ state is empty. The probability to excite $n_{qp}$ quasiparticles simultaneously is, then, $[f_{q=0}(1 - f_{q=\pi}) - f_{q=\pi}(1 - f_{q=0})]^{n_{qp}} = (f_{q=0} - f_{q=\pi})^{n_{qp}}$, where $f_q = (e^{(\varepsilon-\mu)/k_B T} - 1)^{-1}$ is the Bose-Einstein distribution function and the chemical potential $\mu$ is determined by the condition $f_{q=0} + f_{q=\pi} = 1$. The resulting heating rate is

$$\Phi(\Omega) = \Phi_{T=0}(\Omega) \left[\frac{1}{e^{-\mu/k_B T} - 1} - \frac{1}{e^{(4J-\mu)/k_B T} - 1}\right]^{\frac{\hbar\Omega}{4J} + \frac{1}{2}}, \quad (22)$$

where $\Phi_{T=0}$ is given in Eq. (21). As shown in Fig. 4, Eq. (22) (dashed lines) agrees well with the numerical solution for a wide range of temperatures. At very large

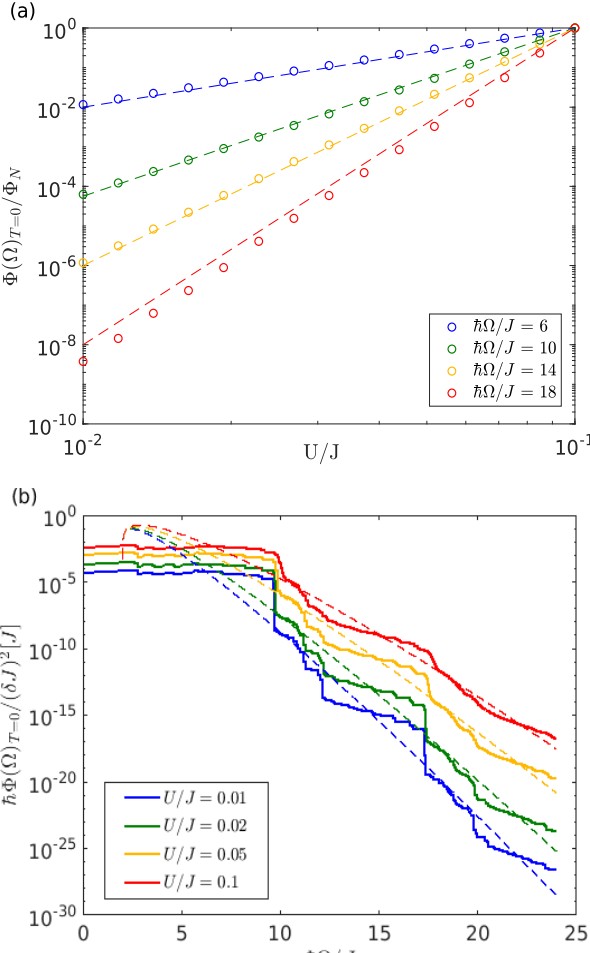

FIG. 3. (a) Heating rate (normalized by $\Phi_N \equiv \Phi(U/J = 0.1)$) as a function of $U/J$ for different values of $\Omega$, obtained from the ED of the Bose-Hubbard model for $N = L = 9$. The numerical results are compared with Eq. 21, which is the analytical continuation of the heating rate for all frequencies (dashed lines). Each range of energies is corresponded to a specific resonance condition, where increasing the external drive requires an additional order in the perturbative expansion of the heating rate. From the linear slope it is evident that for each such range, $\Phi$ scales like $U^2$. In particular, the blue circles corresponded to the regime of $0 < \hbar\Omega/J < 8$, and further increasing the external drive in an additional $4\hbar\Omega/J$, yields the power-law dependent for the other regions. (b) The scaling of the heating rate, Eq. (21) as a function of the external drive. This expression (dashed lines) gives a good estimation for the exponential suppression as obtained numerically from ED.

temperatures, when $k_B T/J$ approaches $J/U$, the subleading orders of our perturbative approach become nonnegligible and the analytical expression deviates from the exact numerical results. Note that as the temperature increases, the thermal weight in the square brackets of Eq. (22) goes to zero. Consequently, the zero-temperature expression Eq. (21) provides an upper

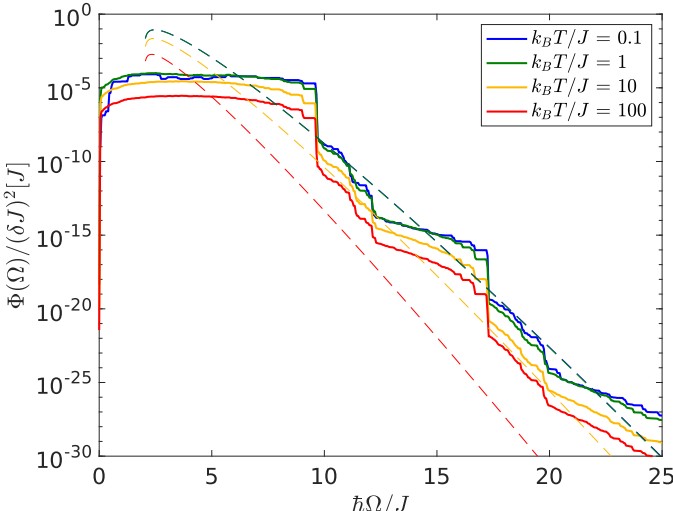

FIG. 4. Temperature dependence of the heating rate, as obtained by ED ($N = L = 9$), compared with the analytical approximation (dashed lines), Eq. (22) for $U/J = 0.01$. The low temperatures curves ($k_B T/J < 0.1$) are almost indistinguishable, and thus not shown explicitly. Our semi-classical approach captures well the exponential decay of the heating rate for a wide range of temperatures, up to $k_B T/J \gtrsim J/U$.

bound for the exponential suppression, which persists at all temperatures.

**Conclusion** – To summarize, we discussed the differences between rigorous [4–8] and statistical [21] Floquet prethermalization. The former approach relies on the boundedness of quantum operators and applies to spin models only. See also Ref. [35], where it was shown that the rigorous approach applied to systems of interacting particles with an unbounded spectrum does not lead to exponential bounds on diffusion rates. The latter approach relies on the statistical description of the prethermal state and applies to a wider range of models, including interacting particles in a lattice and in the continuum [36]. A key difference between these two approaches is that, while the rigorous approach is independent on the initial state, the statistical approach depends on the initial state, through its (quasi)conserved quantities, such as energy and particles' number.

In this article, we applied the statistical argument to the periodically driven Bose-Hubbard model, which was recently realized experimentally [26]. We developed two semiclassical descriptions of Floquet prethermal states, valid in two extreme regimes. The first limit corresponds to strong interactions and large temperatures ($U > k_B T \gg J$), where the suppressed heating rate is the outcome of the low probability to find many particles on a single site. The second limit corresponds to low temperatures and weak interactions ($k_B T < U \ll J$) and is relevant to the experiment of Ref. [26]. Here, the exponential suppression results from the low probability to create simultaneously many quasiparticles in momentum space. In both limits, we described the system semiclassically and applied statistical arguments to derive an analytical expressions for the heating rate $\Phi$ as a function of the driving frequency $\Omega$ and of the temperature $T$. These expressions are found to match the results of the exact diagonalization of the model, without any fitting parameter. Importantly, we demonstrated that in both regimes, the exponential suppression of the heating persists at all temperatures.

In this aspect, the Bose-Hubbard model differs from the coupled rotors model of Refs. [16–21], where the exponential suppression of heating disappears at large temperatures, eventually leading to a runaway from the prethermal regime. This fundamental difference stems from the nature of the conserved quantities of the two models: In the rotor model, the conserved quantity, namely the momentum of the rotors $p_i$, is a continuous variable and can acquire both positive and negative values. At large temperatures, the fluctuations of $p_i$ diverge making the exponential suppression of heating ineffective. In contrast, in the Bose-Hubbard model, the conserved quantity, namely the particles' number $n_i$, is non-negative. If the expectation value of $n_i$ is kept fixed, the fluctuations of this quantity remain finite and the heating rate is suppressed at all temperatures. The prediction of the two models coincide when the average number of particles per site is taken to infinity ($\bar{n} \to \infty$).

Our semiclassical approach disregards effects associated with quantum coherence. In the case of a single kicked rotor, quantum coherence strongly suppresses heating through the dynamical localization in energy space [37, 38]. Accordingly, it was recently shown that dynamical localization can lead to ergodicity breaking in many-body kicked models, such as coupled rotors [39] and the Bose-Hubbard model [40]. However, as conjectured in Ref. [41], dynamical localization is probably restricted to kicked models and, hence, is not relevant to the present study, where we considered a sinusoidal time dependence.

We thank Jonathan Ruhman, François Huveneers, and the authors of Ref. [26] for useful discussions. This work was supported by the Israel Science Foundation, Grants No. 151/19 and 154/19.

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

## Appendix

### A. Matlab script used to plot the semiclassical approximation in Fig. 1

```
1   close all; clear all
2   syms n; syms U;syms mu
3
4   %Temperature is set to one
5   H = U*n^2/2 + mu*n
6   myUs=logspace(-log10(100),-log10(0.01),5)
7   %myUs=logspace(-3,0,4)
8
9   %myUs=myUs(length(myUs):-1:1)
10
11  Nmax=60;
12
13  PP=zeros(length(myUs),Nmax*2/3);
14
15  for u=1:length(myUs)
16      myU=myUs(u)
17      Z = @(mymu) sum(double(subs(exp(-subs(subs(H,U,myU),mu,mymu)),n,0:Nmax)));
18      avn = @(mymu) sum(double(subs(n*exp(-subs(subs(H,U,myU),mu,mymu)),n,0:Nmax)));
19      avn2 = @(mymu) sum(double(subs(n^2*exp(-subs(subs(H,U,myU),mu,mymu)),n,0:Nmax)))
            ;
20      eqn = @(mymu) avn(mymu)/Z(mymu)-1;
21
22      mymu = fzero(eqn,1)
23      myavn2(u)= avn2(mymu)/Z(mymu)
24      P=exp(-subs(subs(H,U,myU),mu,mymu));
25
26      figure(2)
27      semilogy(subs(P,n,0:Nmax));
28      hold on
29      mylegend{u}=['k_BT/U=',num2str(1/myU)];
30
31      allP=double(subs(exp(-subs(subs(H,U,myU),mu,mymu)),n,0:Nmax))/Z(mymu);
32
33      for nOmega=1:(Nmax*2/3)
34          nn=nOmega:Nmax;
35          Pplus=sum(allP(1+nn).*allP(1+nn-nOmega+1));
36          %add 1 because the first item of allP corresponds to n=0;
37          nn=(nOmega+1):Nmax;
38          Pminus=sum(allP(1+nn).*allP(1+nn-nOmega-1));
39          PP(u,nOmega)=2*(Pplus-Pminus);
40      end
```

```
41
42        figure (3)
43        semilogy ([0 ,1:( Nmax∗2/3)] ,[0 ,PP(u ,:)/myU.∗(1:Nmax∗2/3)] , 'linewidth' ,1.0 , 'marker'
              ,'.' , 'markersize' ,15.0)
44        hold on;
45  end
46
47  save('PP.mat' , 'PP' , 'myUs');
48
49  nn=0:Nmax;
50  plot (nn, nn.^2.∗ exp(−log(2)∗nn)/3, 'k—', 'linewidth' ,1.0 , 'marker' ,'.' , 'markersize'
          ,15.0);
51  mylegend{u+1}='Eq. (13)';
52  rubio %plots the inset of Fig. 7 of Ref. [22] (v2)
53
54  xlabel('${\it \Omega^~[U/\hbar]}$', 'Interpreter', 'latex');
55  ylabel('$\Phi{\it k_B T^~[U^2/\hbar]}$', 'Interpreter', 'latex');
56  xlim([0,45]); ylim([1E−8,1]);
57
58  set (gca , 'fontname' , 'times');
59  set (gca , 'fontsize' ,12);
60  set (gca , 'xtick' ,0:10:50);
61  set (gca , 'ytick' ,10.^(−8:2:0));
62        legend (mylegend , 'location' , 'northeast' , 'box' , 'off' , 'fontsize' ,10)
63
64  set (gcf , 'color' , 'white');
65  set (gcf , 'position' ,[100 100 500 300]);
66  saveas (gcf , 'numerics.eps' , 'epsc')
```

### B. Matlab symbolic script used to derive Eqs. (11), (12), and (14)

```
1  syms a; syms n; syms nOmega
2  assume (a>0 & a<1)
3
4  symsum (a^n,n,0 , Inf)
5  simplify((1−a)^2∗symsum(a^(2∗n–nOmega+1) ,n ,nOmega, Inf))
6  simplify((1−a)^2∗symsum(a^(2∗n–nOmega−1) ,n ,nOmega+1, Inf))
7  Pplus=(1−a)^2∗symsum((n^2+(n–nOmega+1)^2)∗a^(2∗n–nOmega+1) ,n ,nOmega, Inf)
8  Pminus=(1−a)^2∗symsum((n^2+(n–nOmega−1)^2)∗a^(2∗n–nOmega−1) ,n ,nOmega+1, Inf)
9  simplify (Pplus–Pminus)
```

### C. Python script used to plot the exact diagonalization in Fig. 1

```
1  #Study the temperature dependence of the heating rate in the Bose–Hubbard model
2
3  from __future__ import print_function , division
4  import sys , os
5  import scipy.io as spio
6  import seaborn as sns
7
8  from quspin.operators import hamiltonian # Hamiltonians and operators
9  from quspin.operators import quantum_LinearOperator # operators
10 from quspin.basis import boson_basis_1d # bosonic Hilbert space
11 import time
12 import numpy as np # general math functions
13 import matplotlib.pyplot as plt # plotting library
14 #
```

```python
15  #plt.rcParams["font.family"] = "Times New Roman"
16
17  ##### define model parameters
18  # initial seed for random number generator
19  np.random.seed(0) # seed is 0 to produce plots from QuSpin2 paper
20  # setting up parameters of simulation
21  L = 8 # length of chain
22  N = L # number of sites
23  nb = 1 # density of bosons
24  sps = L+1 # number of states per site
25
26  J_par = 0.05; U = 1.0; gamma=2*J_par; Emax=9.5; Nomega=200;mylim=[1e-10,2];allT
        =[100,10,1,0.1,0.01];unit='U';PBC=False
27
28  plt.figure(1,figsize=(5,5))
29  sp=sns.color_palette('jet_r',5);#dark#rainbox
30  plt.subplot(212)
31
32  #### Numerics: Diagonalizing Hamiltonian
33  filename = "/data/ED3_JoU"+str(J_par/U)+"_N"+str(N)+"_PBC"+str(PBC)
34  if not os.path.exists(filename+"allE.npy") :
35      print("Running",filename)
36      tic=time.time()
37      ##### set up Hamiltonian and observables
38      int_list_1 = [[-0.5*U,i] for i in range(N)] # interaction $-U/2 \sum_i n_i$
39      int_list_2 = [[0.5*U,i,i] for i in range(N)] # interaction: $U/2 \num_i n_i^2$
40      if PBC :
41          hop_list = [[-J_par,i,(i+1)%N] for i in range(0,N,1)] # PBC
42      else :
43          hop_list = [[-J_par,i,i+1] for i in range(0,N-1,1)] # OBC
44      hop_list_hc = [[J.conjugate(),i,j] for J,i,j in hop_list] # add h.c. terms
45      # set up static and dynamic lists
46      static = [
47                  ["+-",hop_list], # hopping
48                  ["-+",hop_list_hc], # hopping h.c.
49                  ["nn",int_list_2], # U n_i^2
50                  ["n",int_list_1] # -U n_i
51              ]
52
53      #Note that "perturbation" is proportional to J_par --> need to devide Phi by
            J_par**2
54      perturbation =      [
55                  ["+-",hop_list], # hopping
56                  ["-+",hop_list_hc] # hopping h.c.
57              ]
58      dynamic = [] # no dynamic operators
59
60      basis = boson_basis_1d(N,nb=nb,sps=sps)
61      print("total H-space size: {}".format(basis.Ns))
62
63      H_BHM = hamiltonian(static,dynamic,basis=basis,dtype=np.complex128)
64      allE,allV=H_BHM.eigh()
65      hop=hamiltonian(perturbation,dynamic,basis=basis,dtype=np.complex128)
66      matrix_elem2=np.power(np.abs((hop.rotate_by(allV,generator=False)).toarray()),2)
            ;
67
```

```python
68        np.save(filename+"allE.npy",allE)
69        np.save(filename+"me.npy",matrix_elem2)
70        toc=time.time();print("Time : ",toc-tic)
71
72    else :
73        print('Loading',filename)
74        allE=np.load(filename+"allE.npy")
75        matrix_elem2=np.load(filename+"me.npy")
76
77
78    #### Numerics: Computing the spectrum
79    allomega=np.linspace(0,Emax,Nomega);
80
81    for c in range(len(allT)):
82
83        T=allT[c]
84        tic=time.time()
85        filename2 = filename+"_T"+str(T)+"_gamma"+str(gamma)+"_Emax"+str(Emax)+"_Nomega"
                +str(Nomega)
86
87        if not os.path.isfile(filename2+".npy") :
88            print('Running',filename2)
89            allPhi=np.zeros(Nomega);
90
91            Ej,Ek = np.meshgrid(allE,allE);
92
93            Z=np.sum(np.exp(-(allE-allE[0])/T));
94            Pj = np.exp(-(Ej-allE[0])/T)/Z;
95            Pk = np.exp(-(Ek-allE[0])/T)/Z;
96
97            P0 = (Pk-Pj)*matrix_elem2
98            for w in range(Nomega) :
99                deltaE=Ej-Ek-allomega[w]
100                allPhi[w]=np.sum((P0*(deltaE<gamma)*(deltaE>-gamma)/gamma/2))
101
102            toc=time.time()
103            print("Time:",toc-tic)
104            np.save(filename2+".npy",allPhi);
105        else :
106            print('Loading',filename2)
107            allPhi=np.load(filename2+".npy");
108
109        plt.semilogy(allomega,T*allPhi/(1e-15+allomega)/J_par**2/L/2,label="T/U= "+str(T
                ),color=sp[c]);#sp(colori[c]));
110
111
112    #### Semiclassical approximation (laoding from Matlab)
113    mat = spio.loadmat('../PP.mat', squeeze_me=True)
114    PP=mat['PP']
115    myUs=mat['myUs']
116    print('Loaded theory for T/U',1/myUs)
117    sh=PP.shape
118    Nmax=sh[1];
119
120    for s in [1,2]:
121        ax=plt.subplot(210+s)
```

```python
122        nn=np.array(range(1,45))
123        plt.semilogy(nn,1/3*np.power(1/2,nn/U)/2/gamma,'k.:',label="Eq. (13)")
124        nn=np.array(range(1,Nmax+1))
125        for i in range(sh[0]) :
126            plt.semilogy(nn,PP[i,:]/myUs[i]/nn/2/gamma,'.:',label=r'$k_B T/'+unit+'='+
                   str(1/myUs[i])+"$",color=sp[i])#sp(colori[i]));
127        plt.xlabel(r"$\Omega\ ["+unit+"/\hbar]$");
128        plt.ylabel(r"$ \left(\Phi k_B T\right)\ /\ (\Delta J^2\ \Omega)\ [\hbar/"+unit+"
                   ]$");
129        box = ax.get_position()
130        print(box)
131        plt.ylim(mylim);
132        ax.set_position([box.x0+0.05*box.width, box.y0+0.05*box.height*(3-s), box.width,
                   0.95*box.height])
133        plt.yticks([1e-10,1e-8,1e-6,1e-4,1e-2,1])
134
135 plt.subplot(211)
136 plt.xlim([0,Nmax+5]);
137 plt.legend(loc=1)
138 plt.subplot(212)
139 plt.xlim([0,Emax]);
140 plt.savefig("../"+filename[5:]+".pdf")
141 plt.show()
```

## D. Finite size effects

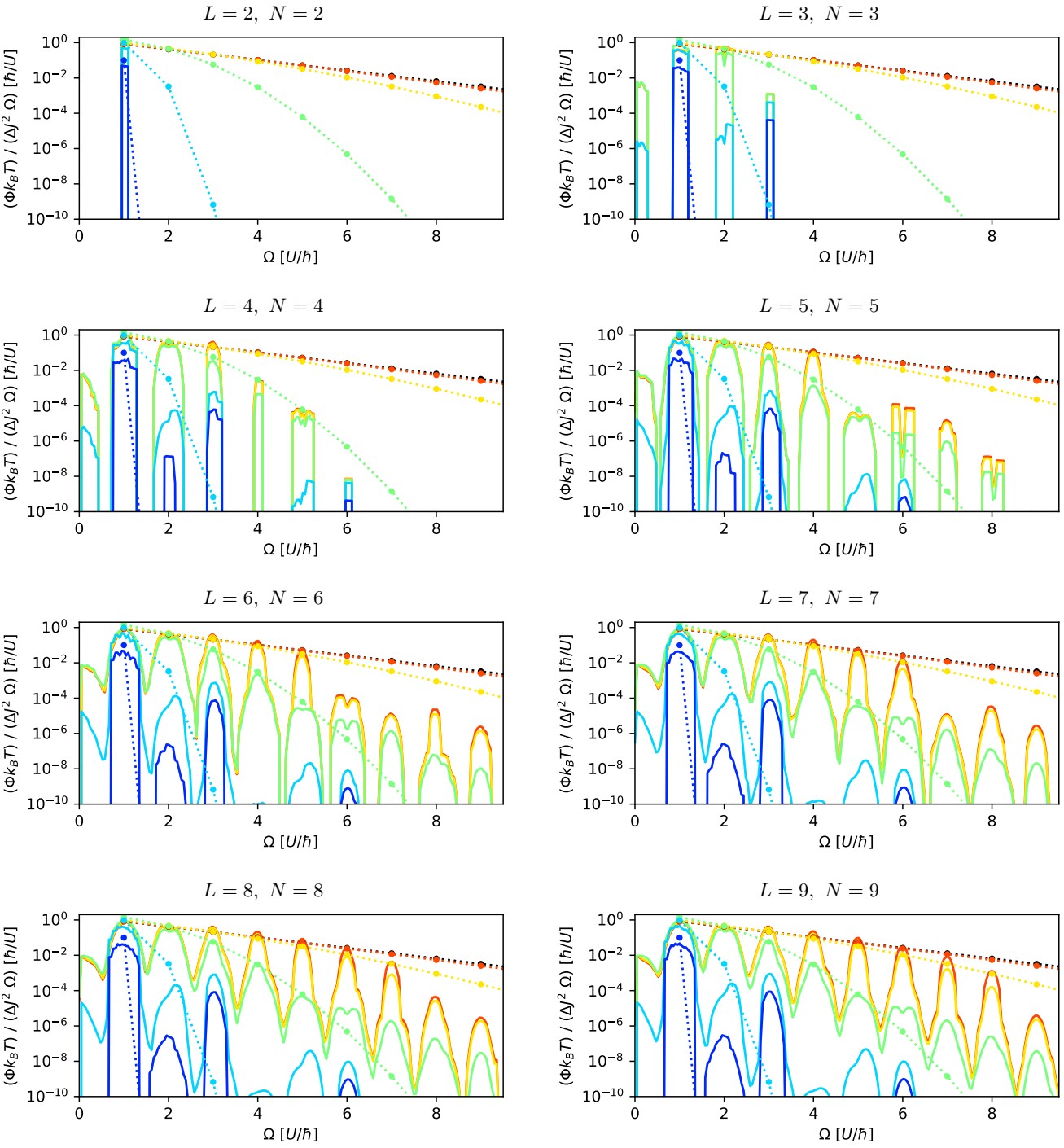

FIG. 5. Same as Fig. 1 for $N$ particles on $L$ sites. No fitting parameter is used. The semiclassical approximation matches the exact results for frequencies $\hbar\Omega < N/U$ and temperatures $k_B T > U$.