# Peer review of "Statistical Floquet prethermalization of the Bose-Hubbard model"

_SciPost Physics Core_

## Round 4 · Referee Report · Anonymous (Referee 1) · 2020-7-12

Strengths

The paper gives a clear explanation of the exponential suppression of heating rate with driving frequency for systems whose Hamiltonian is not bounded by an energy scale J. Well ... given the presence of number conservation, the local terms of the Hamiltonian are actually bounded, but 1) not in the thermodynamic limit and 2) from Eq.(1) one would not be able to see the 1/T dependence of the heating rate.
So this paper brings very interesting insight into the problem and significantly pushes forward the field.

Weaknesses

Fig.1 is not well described.

The explanation of why a high-temperature analysis is relevant is missing.

A more thorough overview of the field of periodically driven systems could be given (see Report).

Report

I have read with great interest this nice work by Dalla Torre. The paper is fairly well written and the results are interesting and relevant to most current theoretical and experimental research.
The paper gives a clear explanation of the exponential suppression of heating rate with driving frequency for systems whose Hamiltonian is not bounded by an energy scale J. Well ... given the presence of number conservation, the local terms of the Hamiltonian are actually bounded, but 1) not in the thermodynamic limit and 2) from Eq.(1) one would not be able to see the 1/T dependence of the heating rate.

This paper follows fairly naturally from [21], however it is brings enough novelty and insight to deserve a high profile publication.

There are some clarifications I would like to have though:

1) Is an high temperature T analysis relevant to the experiment [23] ?
Is it because the driving first brings the state far from low temperature and then, due to what described in this work, the suppression of heating occurs?

2) Fig.1 is a bit messy:
The caption is not clear. What are the two different panels about? Is panel (b) just an enlarged version of panel (a)? The why there seems to be used 2 different values of J/U? An why in the legend you write Eq.(13)? I think it would be (15) now. Maybe the figure was prepared when Eq.(15) was actually in position (13).
Also it is not clear where the continuous lines come from. Are they derived from (16)?
Furthermore the caption contains 2 typos:
- Caption of Fig.1 -> J/U = 0_05. instead of 0.05 I think ...
- Also in the same caption Eq. (15 no closing parenthesis

3) The suppression observed in Fig.1 seems to be larger than exponential. Any understanding of that?

4) By reading the literature on periodically driven systems, and heating, I cannot avoid incurring in works of A. Eckardt whom however is not cited at all in this work. It is quite surprisiping to me as, for example, he is a co-author also of this experimental work PRL 119, 200402 (2017), and the author of an important review in the field.

5) To give a more comprehensive picture of the field, I also find relevant to mention the works PHYSICAL REVIEW E 97, 022202 (2018), PHYSICAL REVIEW B 101, 064302 (2020) and related ones. This would help a reader.

6) until before Eq.(5) you have hbar and then in 9 you don't. Also the units for Phi should be energy over hbar. Since delta function has units of 1/energy, maybe all you need is to divide the right-hand side by hbar. Overall the use of hbar is a little inconsistent.

7) This is a comment to improve the clarity of the derivation. The author considers corrections to \delta(\Omega - \Delta E) due to tunneling J. Can we have an idea of why only using the corrections in this function compared to the probabilities P in Eq.(8) is a reasonable approach? I think this could help the reader.

8) I am not sure about the exponent in Eq.(14) ... is it \hbar\Omega/U maybe?

9) There may be a factor 1/2 in Eq.(15) coming from the +1 in z^{\Omega/U +1}, but maybe I am wrong.

some typos
- Page 1, second column -> "This effect was explainED in Ref. [21]"
- Put together [23] [24].
- "more than one particleS" remove s (I think)

Requested changes

See Report

---

## Round 4 · Referee Report · Anonymous (Referee 2) · 2020-9-28

Strengths

1)relevant topic (theoretical and experimental interest)
2)nice results obtained with relatively simple techniques - clear physical picture
3)comparison between analytics and exact numerics

Weaknesses

1)presentation of the results and of the assumptions behind the theory could be improved (see requested changes)

Report

This work focuses on interacting lattice bosons, described by the Bose-Hubbard mode, driven by a periodic hopping modulation. This is a problem relevant to recent cold atoms experiments and to the field of Floquet quantum many body systems.

The author discusses the linear-response heating rate of the system for high temperatures and large interactions and its dependence from drive frequency. Analytical (statistical) arguments are compared with exact numerical calculations.
The results show that, also for this model with unbounded local Hilbert space, the linear response heating rate of the system is exponentially suppressed in drive frequency, with a temperature dependent prefactor (at least in the high-T regime considered here).

I think this is an interesting and topical work that deserves publication in SciPost Physics Core. I nevertheless find that the current manuscript could benefit from a better clarification of the points below:

Requested changes

1)The basic idea of statistical floquet prethermalization (Eq 2, Eq. 8) is that the prethermal state is described by a Gibbs state of the Floquet Hamiltonian. In practice, in this work, how is the temperature T evaluated? It seems to me the author uses it as an external parameter, or identifies it as the initial temperature (before the drive is switched on). What is the rationale behind this? It would be useful to further elaborate on this point.

2) The paper focuses on the linear response heating rate and uses perturbation theory in the drive amplitude to obtain both Eq 9 and Eq16.
What happens beyond this regime? Can this statistical approach be generalised? Does the main result of this work still holds? I would expect that for strong drive amplitude the system would be still able to absorb energy even when Omega/U=n>>1 through non-perturbative effects (see also point #5)
It would be good if the author could comment in the manuscript on this point and also mention clearly the regime of validity of the present analysis.

3)The comparison with ED shown in the bottom panel of Figure 1 reveals some oscillations of the heating rate which are absent in the statistical approach. Is it clear where do these come from?I suspect they might come from a mechanism of "resonant thermalisation", see point below.

4) In the same figure, I find particularly interesting the deviations at small drive frequencies, discussed in the paper towards its end and attributed to quantum resonances- beyond the current semiclassical approximation- and leading to higher heating rates.
This effect has been discussed for example in Phys. Rev. Lett. 120, 197601 (2018) in the context of Fermi-Hubbard model at large interaction, where it has been shown that sweeping the drive frequency across the condition Omega=U (or multiples) could lead to a rapid "resonant" thermalisation and increased heating. The numerical results of this paper seems to indicate a similar mechanism also in the Bose Hubbard, which I think is very interesting.

5)Related point: given this "resonant" condition, I would then suspect that increasing enough the drive amplitude deltaJ could result in an increased heating rates even at large (resonant) frequencies Omega/U=n>>1...Could the author comment on this point?

6)The caption of Figure 1 could be improved, referring to the separate panels and clarifying what has been obtained with the analytical theory and what with exact numerics.

---

## Editorial Decision

resubmitted